# Stealth Liposomes Encapsulating a Potent ACAT1/SOAT1 Inhibitor F12511: Pharmacokinetic, Biodistribution, and Toxicity Studies in Wild-Type Mice and Efficacy Studies in Triple Transgenic Alzheimer’s Disease Mice

**DOI:** 10.3390/ijms241311013

**Published:** 2023-07-02

**Authors:** Adrianna L. De La Torre, Thao N. Huynh, Catherine C. Y. Chang, Darcy B. Pooler, Dylan B. Ness, Lionel D. Lewis, Sanjana Pannem, Yichen Feng, Kimberley S. Samkoe, William F. Hickey, Ta Yuan Chang

**Affiliations:** 1Department of Biochemistry and Cell Biology, Geisel School of Medicine at Dartmouth, Hanover, NH 03755, USA; adrianna.delatorre.gr@dartmouth.edu (A.L.D.L.T.);; 2Clinical Pharmacology Shared Resource, Norris Cotton Cancer Center, Dartmouth-Hitchcock Medical Center, Lebanon, NH 03766, USA; 3Thayer School of Engineering, Dartmouth College, Hanover, NH 03755, USA; sanjana.pannem.th@dartmouth.edu (S.P.); yichen.feng.gr@dartmouth.edu (Y.F.);; 4Department of Pathology, Dartmouth-Hitchcock Medical Center, Lebanon, NH 03766, USA; william.f.hickey@dartmouth.edu

**Keywords:** ACAT1/SOAT1, nanoparticles, Alzheimer’s disease, cholesterol, cholesteryl ester, amyloid, tau, F12511, DSPE-PEG, phosphatidylcholine

## Abstract

Cholesterol is essential for cellular function and is stored as cholesteryl esters (CEs). CEs biosynthesis is catalyzed by the enzymes acyl-CoA:cholesterol acyltransferase 1 and 2 (ACAT1 and ACAT2), with ACAT1 being the primary isoenzyme in most cells in humans. In Alzheimer’s Disease, CEs accumulate in vulnerable brain regions. Therefore, ACATs may be promising targets for treating AD. F12511 is a high-affinity ACAT1 inhibitor that has passed phase 1 safety tests for antiatherosclerosis. Previously, we developed a nanoparticle system to encapsulate a large concentration of F12511 into a stealth liposome (DSPE-PEG_2000_ with phosphatidylcholine). Here, we injected the nanoparticle encapsulated F12511 (nanoparticle F) intravenously (IV) in wild-type mice and performed an HPLC/MS/MS analysis and ACAT enzyme activity measurement. The results demonstrated that F12511 was present within the mouse brain after a single IV but did not overaccumulate in the brain or other tissues after repeated IVs. A histological examination showed that F12511 did not cause overt neurological or systemic toxicity. We then showed that a 2-week IV delivery of nanoparticle F to aging 3xTg AD mice ameliorated amyloidopathy, reduced hyperphosphorylated tau and nonphosphorylated tau, and reduced neuroinflammation. This work lays the foundation for nanoparticle F to be used as a possible therapy for AD and other neurodegenerative diseases.

## 1. Introduction

Alzheimer’s disease (AD) is the most common form of dementia. Arguably, there is no therapy to slow the clinical progression of the disease. For AD disease modifying therapies, monoclonal antibodies targeting amyloid have advanced to the final clinical trial stage, with aducanumab and lecanemab being the drugs that were most recently approved by the Food and Drug Administration (FDA) [1]. Monoclonal antibodies against tau, as well as against certain inflammatory markers, are also in the clinical pipeline. Besides antibodies, the conventional use of small molecules comes with challenges in crossing the blood–brain barrier (BBB) (~98% of small molecules are not able to cross the BBB) [2], as well as possible off-target effects and toxicity. However, one of the key benefits of small molecules is their relatively simple manufacturing and cost effectiveness when compared to biologics. With a disease as complex as AD, it is likely that a cocktail of therapies will be needed to slow the progression and ultimately cure the disease.

AD is classified as early onset (EOAD) or late onset (LOAD), with 99% being LOAD. AD pathological hallmarks consist of extracellular amyloid plaques composed of amyloid beta peptides and neurofibrillary tangles composed of misfolded and hyperphosphorylated tau. Lipid granules also accumulate within the glia, as reviewed in [3]. LOAD involves multiple genetic risk factors; among them, *APOE* ε4, *CLU*, *ABCA7*, and *ABCA1* are all involved in lipid metabolism [4,5,6,7]. Thus, AD can be considered as a special lipid disease. 

Cellular cholesterol homeostasis is under tight control mechanisms that are involved in the uptake of exogenous cholesterol-rich substances, biosynthesis of endogenous cholesterol, intracellular transport, and efflux of cholesterol, and the excess cellular cholesterol is stored as cholesterol esters (CEs) (for reviews see [8,9,10,11]). Normally, CE levels in the brain are very low. However, in the vulnerable regions of brain samples with LOAD, CE levels arose by 1.8 fold [12]. Similar results were observed in three EOAD mouse models [12,13]. Additional studies showed that in AD patient-derived neurons, an increase in CE contents correlates positively with the tau pathology [14]. CEs are biosynthesized by the enzymes acyl-coenzyme A:cholesterol acyltransferases (abbreviated as ACATs) (also named sterol O-acyltransferases (SOATs) in GenBank. In GenBank ACATs are assigned to acetyl-CoA acetyltransferases, which are separate enzymes that produce acetyl CoA, but not CEs). There are two *ACAT*/*SOAT* genes that encode two homologous enzymes [15,16,17,18]. Both enzymes use long-chain fatty acyl-CoAs and sterols as substrates [19]. ACAT1 is expressed in essentially all cells, including cells in the brain, and ACAT2 is mainly expressed in intestinal enterocytes and hepatocytes. In various cell types in the CNS, the gene expression levels of *ACAT1* are much higher than those of *ACAT2* (data retrieved from the public domain can be found in [20]). ACAT1 and ACAT2 are membrane proteins located in the endoplasmic reticulum (ER), and both enzymes are allosterically activated by sterols [21,22,23,24]. In addition, in microglia isolated from various neurodegenerative diseases and in vulnerable regions of human brains from LOAD, the *ACAT1/SOAT1* gene is modestly induced (data retrieved from public domain can be found in [20]). 

At the preclinical level, evidence from several laboratories implicates ACAT1 as an important molecular target for the treatment of AD [25,26,27,28,29,30,31]. Mechanistically, an ACAT1 blockade by small molecule inhibition or by genetic inactivation (A1B) was reported to offer multiple benefits to AD models including the following: (1) A1B increased the content of the neuroprotective oxysterol 24(S)-hydroxycholesterol in the AD mouse brain [27] and in the AD patient iPSC-derived human neurons [30]. (2) A1B increased the autophagy flux and led to the clearance of Aβ oligomers in microglia [29], and it also increased the clearance of misfolded tau in neurons [32,33]. (3) In EOAD patient-derived neurons, A1B reduced the CE content and prevented their inhibitory effects on tau proteostasis [30]. (4) A1B decreased the protein content of mutant full-length hAPP in the brain of an EOAD mouse model [27] and of iPSC-derived human neurons in AD patient [30]. (5) A1B cleared the CEs that accumulated in myelin-debris-treated microglia that lacked TREM-2, which is a risk factor for LOAD [31]. Genetic variants of human *Acat1* have been studied by several research groups. To cite one example, among the four common *ACAT1* single-nucleotide polymorphism (SNP) investigated, one exhibits a protective haplotype and one exhibits a risk haplotype for dementia development [34]. In addition, at least one human who has a putative homozygous knockout mutation for *SOAT1* had been identified without obviously noticeable issues [35], supporting the idea that A1B may not cause overt toxicities in humans. Regarding small-molecule ACAT inhibitors, ACAT is a drug target that is used to treat atherosclerosis, and many ACAT inhibitors were produced as antiatherosclerosis agents. Three of these inhibitors, Avasimibe—2,6-bis(1-methylethyl)phenyl ((2,4,6-tris(1-methylethyl)phenyl)acetyl)sulfamate (CI1011), 2-[4-[2-(Benzimidazol-2-ylthio)ethyl]piperazin-1yl]-N-[2,4-bis(methylthio)-6-methyl-3-pyridyl]acetamide dihydrochloride, 4-[2-(1H-Benzimidazol-2-ylthio)ethyl]-N-[6-methyl-2,4-bis(methylthio)-3-pyridinyl]-1-piperazineacetamide dihydrochloride (K604), and Eflucimibe—(S)-2′,3′,5′-trimethyl-4′-hydroxy-alpha-dodecylthio-alpha-phenylacetanilide (F12511), had passed clinical phase 1 safety tests [36]. They were all abandoned, some of them (CI1011) for their lack of efficacy to act as a supplement to statin to further reduce serum cholesterol levels [37]; F12511 or K604 were also withdrawn from clinical trials without a provided reason. CI1011 is a weak ACAT inhibitor (Ki = 20 µM for both ACAT1 and ACAT2) [38]. K604 is a high-affinity ACAT1-specific inhibitor with Ki = 0.45 µM [38]. F12511 is a high-affinity inhibitor of ACAT1 (Ki = 0.039 µM); it also inhibits ACAT2 but with less potency (Ki = 0.110 µM) [39]. CP113,818 is a high-affinity ACAT inhibitor [40] (Ki = 0.02 µM for both ACAT1 and ACAT2) [39]. When CP113,818 was delivered by implanting as part of a custom designed slow-releasing pellet underneath the mouse skin, it was effective in suppressing amyloidopathy in a mouse model for EOAD [26]. This result strongly suggests that ACAT inhibitors can be effective to treat AD. Unfortunately, to our knowledge, the composition of the slow-releasing pellet remained as a company proprietary information. In addition, as an antiatherosclerosis drug candidate, CP113,818 had failed during the preclinical trial because it accumulated in adrenals in animals and caused toxicity. It is not clear what causes CP113,818 to be toxic. CP113,818 possesses an asymmetric carbon, and this asymmetry is needed for CP113,818 to act as an ACAT inhibitor. The enantiomeric isomer of CP113,818 (and its closely related structural analogs) is inactive as an ACAT inhibitor, but it also causes severe adrenal toxicity [41,42], suggesting that the toxicity of CP113,818 and its close structural analogs such as ATR-101 [43,44] is mainly due to their physicochemical characteristics that are unrelated to their ability to inhibit ACAT [41,42]. Supporting this interpretation is the evidence that the adrenal functions of mice genetically knocked out of *Acat1* is normal [45]. To overcome adrenal toxicity by CP113,818, Pierre Fabre pharmaceutical produced F12511 (abbreviated as “F”). F12511 also contains an asymmetric center (the chemical structures of CP 113,818 vs. F12511 can be found in [46]) and is a high-affinity inhibitor of ACAT1 (Ki = 0.039 µM). Unlike CP113,818, F12511 passed phase 1 safety tests for antiatherosclerosis compounds [36].

The other high-affinity ACAT1 inhibitor, K604 (the chemical structure of K604 can be found in [46]), has also passed phase 1 safety tests for antiatherosclerosis in humans. K604 is known to be impermeable to the mouse brain [47]. However, whether F12511 is permeable to the brain is unknown. Here, we encapsulated F12511 or K604 in the DSPE-PEG_2000_/PC mixed liposome [48] and delivered each to wild-type (WT) mice by IV injections and performed an HPLC/MS/MS analysis and ACAT enzyme activity measurement. We also performed a 2-week IV delivery of nanoparticle F to aging 3xTg AD mice to determine if nanoparticle F can ameliorate amyloidopathy, tauopathy, and neuroinflammation, the three major biomarkers for LOAD. The results are reported in the current manuscript.

## 2. Results

### 2.1. F12511 Is Present in the Brain after Single IV Injection of Nanoparticle F

In order to determine an optimal formulation to inhibit ACAT in the brain, we encapsulated F12511 or K604 as part of a DSPE-PEG_2000_-based nanoparticle (also called a “stealth liposome”; when delivered to the blood, DSPE-PEG_2000_ protects the liposome from rapid hepatic degradation in vivo). Nanoparticles were prepared based on the procedure described in [48]. We previously found that when the high-affinity ACAT inhibitor such as F12511 or K604 are added to cells, the inhibition of the ACAT enzyme by F12511 or K604 remains unaffected by cell washing or the cell homogenization preparation process [24,48]. This finding implicates that once the inhibitor binds to the ACAT enzyme, it stays firmly bound for a few hours before dissociating from the enzyme. Based on this finding, we treated the animals with the inhibitor by IV injection, perfused the mice after the animals were treated (to remove blood contamination from the tissues), then prepared the tissue homogenates, and monitored the ACAT enzyme activity in these tissues by using the mixed micelles enzyme assay described previously [24,39,49,50] (the mixed micelles assay measures the ACAT enzyme activity independent of endogenous lipid composition because it reconstitutes the ACAT protein in mixed micelles with a defined lipid composition). We injected nanoparticles with F12511 (nanoparticle F) at 5.8 mg/kg or with K604 at 40 mg/kg by IV. Four hours after IV, we sacrificed the mice and compared the ACAT enzyme activity remaining in various tissues in nanoparticle-F- or K604-treated mice vs. that of the untreated mice. The results (Figure 1) showed that after using our nanoparticle system, F12511 inhibited the ACAT enzyme in the brain, as well as in the adrenal glands, while K604 only inhibited the ACAT enzyme in the adrenal glands and not in the brain. This result shows that as part of nanoparticles, F12511, but not K604, is permeable to the mouse brain. This result is consistent with the work of Shibuya et al. [47], who showed that K604, a quite hydrophilic molecule, is impermeable to the brain. This result also shows that even at a relatively low dose (5.8 mg/kg), 8 h after IV injection, F12511 is still able to inhibit ACAT activity in the brain by more than 50%. Overall, our results demonstrate that, in our nanoparticle system, F12511 exhibited significantly stronger inhibitory effects on ACAT activities in the mouse brain compared to K604. Therefore, the subsequent focus of our study will be on understanding the pharmacokinetics, biodistribution, and efficacy of nanoparticle F in the brain.

To determine the extent to whether F12511 alone reaches the brain parenchyma to inhibit ACAT, we conducted a study involving the daily oral gavage administration of nanoparticle F for four consecutive days, and we measured the ACAT activity 6 h after the final administration. Due to the hydrophobic nature of F12511, its delivery in animals at a high dose poses challenges [51]. Previous studies have utilized a delivery vehicle, such as cyclodextrin, to deliver F12511 in animals [52]. In this experiment, we employed an oral gavage administration of the nanoparticle F (Appendix A) as a means of ensuring the high-dose delivery of F12511. It is important to note that liposome-based nanoparticles are expected to degrade in the intestine upon oral delivery [53]. Consequently, F12511 is released into the bloodstream, mimicking the administration of F12511 alone. Our result (Appendix A) revealed a lack of ACAT activity inhibition in the brain even after four daily administrations. However, significant ACAT activity inhibition was observed in peripheral tissues such as the adrenal glands and livers. These findings suggested that without a carrier, the penetration of F12511 into the brain parenchyma is insufficient to inhibit ACAT activity.

To examine if IV injection of nanoparticle F causes the preferential accumulation of F12511 in adrenals or in any other tissues, we conducted mass analyses of nanoparticle F by using a LC/MS/MS-based procedure described in [54]. The results (Figure 2, top) showed that after a single IV injection of nanoparticle F at a high dose (46 mg F12511/kg), at 4 h, the F12511 content in the plasma reached values close to 30 µM. To compare this value with the values in the literature, it was previously known that F12511 can be solubilized at a high concentration by cyclodextrin. When mice were orally fed with an F12511/cyclodextrin complex, the maximal blood F12511 concentration that could be reached was only 0.75 µM (data retrieved from the Pierre Fabre patent US patent 5990173); i.e., a value 30-fold less than the method described here can achieve. The results in the bottom graph of Figure 2 showed that after a single IV of nanoparticle F, the F12511 content decreased rapidly in the plasma, adrenals, livers, and brains with time (from 4 to 24 h); these results, which were collected via mass spectrometry analyses, indicated that there was no evidence for the preferential accumulation of F12511 in the adrenals. We also monitored the ACAT enzyme activities in various tissues 4 to 48 h after a single IV injection of nanoparticle F at 46 mg/kg. The results (Figure 3) showed that similar to the F12511 tissue content analyses results (Figure 2, bottom), the ACAT enzyme activity in various tissues (including the adrenal glands) steadily returned to near-normal levels within 24 to 48 h, demonstrating that there is no evidence to suggest the prolonged, preferential accumulation of F12511 in the adrenals.

### 2.2. IV Injection of Nanoparticle F Does Not Produce Overt Systemic or Neurotoxicities In Vivo

We had previously shown that treating nanoparticle F at very high concentrations to mouse primary neurons in a culture did not produce obvious toxicities [48]. Previous work by others showed that when fed to mice for seven days, certain toxic ACAT inhibitors such as AZD3988 caused a significant reduction in fine vacuolation in cortical regions [55]. Here, we tested if nanoparticle F would produce toxicity in various tissues by giving IV injections of nanoparticle F to mice at 46 mg/kg once per day for 7 days. Two days after the last injection, we sacrificed the mice and examined the morphologies of various brain regions, the liver, and the adrenal cortices after histochemical staining (Figure 4). As shown in Figure 4A,B, no detectable morphological alteration was found in the CNS tissues or in the liver; nanoparticles with or without F might have induced certain small but detectable vacuolization in the adrenal (Figure 4B). The exact cause for this observation is unknown. Adrenal cortical vacuolization in rats and mice [56] is characterized by the accumulation of clear vacuoles within cells, mainly in the zona fasciculata but also in the zonae reticularis and glomerulosa. The vacuolization can be focal or diffuse. We speculate that the vacuoles seen (Figure 4B) may represent accumulations of cholesterol and/or other lipids.

### 2.3. Nanoparticle F Is Present and Detectable in the Brain by Using the Fluorescent Dye DiR

To monitor the biodistribution of the nanoparticles, we adapted the procedure from Bishnoi et al. [57] and Meng et al. [58] by encapsulating the highly fluorescent dye 1,1′-dioctadecyl-3,3,3′,3′-tetramethylindotricarbocyanine iodide (DiR), which is a hydrophobic long-chain dialkylcarbocyanine [58]. DiR is a lipophilic NIR dye that can be incorporated into the lipid bilayer of nanoparticles [58,59]. Moreover, unencapsulated free DiR exhibits only a weak fluorescence signal, which does not interfere with the signals from encapsulated DiR nanoparticle [58]. After the IV delivery, we isolated different mouse brain tissues including tissues from the forebrain and cerebellum to produce tissue homogenates and quantitated the DiR signal present in these homogenates. The result showed that after 4 h of IV injection, the DSPE-PEG/PC/DiR nanoparticles were present in various regions of the brain (Figure 5). The quantitation of the results shown in (Figure 5) suggested that about 0.3–0.5% of the nanoparticles injected into the blood entered the brain interior. This value is consistent with the literature-reported value [60]. However, it can only be considered as semiquantitative because the values reported in Figure 5 are influenced by the tissue environment, which affect the DIR fluorescence signal. To compare the HPLC/MS/MS analyses of F12511, the result showed that at the 12 to 24 h time point, about 0.5% to 1% of F12511 in the plasma was found in the brain interior (Figure 2, table on top). Together, we speculate that soon after the IV injection, a portion of F12511 may dissociate from the nanoparticle and undergo rapid degradation by the liver, while other portions of F12511 might enter the brain along with the nanoparticle.

### 2.4. Nanoparticle F Suppressed Amyloid-Beta (Aβ) and Diminished the Levels of Both Unphosphorylated and Hyperphosphorylated Tau (HPTau)

We had previously shown that *Acat1/Soat1* gene knock out (KO) in the 3xTg AD mouse model [61] reduced the mutant full-length hAPP, suppressed the level of Aβ [27], and diminished the levels of unphosphorylated mutant human Tau, but did not hyperphosphorylate the mutant human Tau (HPTau) [32]. Here we tested the therapeutic efficacy of nanoparticle F in the 3xTg AD mouse model. In this model, Aβ, tau phosphorylation, and neuroinflammation become pronounced between 12–20 months of age [62]. We tested nanoparticle F in mice with an advanced age (16–20 months). Male and female mice were left untreated or were treated with daily IV injections (we alternated between the tail vein and retro-orbital venous sinus; 200 µL per injection) of nanoparticle F (~46 mg/kg F12511) or DSPE-PEG_2000_/PC (nanoparticle alone) for 2 weeks. After the treatment, the mouse tissues were collected and snap frozen until they were ready for tissue homogenization, Western blot analyses, ELISA, and a Luminex assay. The results showed that nanoparticle F significantly reduced the mutant hAPP as determined with the Western blot (Figure 6A) and reduced the total acid extractable Aβ1-42 as determined with the ELISA assay (Figure 6B). Interestingly and unexpectedly, the nanoparticles alone also reduced the full-length hAPP, although the effect was milder than that of the nanoparticles with F12511 (Figure 6A), suggesting that the nanoparticles without F12511 may exert a certain suppressive effect on the full-length hAPP. We noted that, in Figure 6, there was a large variability in the expression levels of the mutant hAPP and the Aβ1-42 levels among the nanoparticle-alone/vehicle (DSPE-PEG/PC)-treated and the untreated groups. We suspect that the variability might have been caused by the variability in the ages of these mice used (16–20 months of age).

Besides the amyloid deposition, the 3xTg AD mice also developed tau pathologies. We measured changes in tau by performing Western blots by using two antibodies: HT7 (which recognizes the total human tau, hTau) and AT8 (which recognizes the human tau phosphorylated at Ser202/Thr205). The results showed that when compared with the untreated mice, only nanoparticle F, but not the nanoparticles alone, significantly reduced the total human tau levels (Figure 7A). Interestingly and unexpectedly, both nanoparticle F and the nanoparticles alone significantly reduced the phosphorylated tau HPTau levels (Figure 7B). These results again suggest that the nanoparticles alone may exert a certain suppressive effect on HPTau levels.

### 2.5. Nanoparticle F Attenuates Neuroinflammation in the Aging 3xTg AD Mice

The aging 3xTg AD mice exhibited chronic neuroinflammation [62]. Here we tested if nanoparticle F may attenuate neuroinflammation in these mice at 16–20 months of age. After a two-week IV injection to these mice, we prepared whole brain homogenates and analyzed 31 different mouse cytokines by using Milliplex^®^ technology [63]. The result in Figure 8A compares the average cytokine expression level between the different experimental treatment groups. Our data suggested that, when compared to the nanoparticle treatment alone, the treatment with nanoparticle F resulted in a decrease in the trend of 27 out of the 31 analyzed cytokines, of which most were proinflammatory in nature (first two columns from the left, Figure 8A). In some cytokines, such as TNF-α and Eotaxin, the nanoparticles and F12511 worked additively to reduce their levels (Figure 8A). To further examine the effect of the nanoparticles and nanoparticle F in the aging 3xTg AD mice, we plotted individual cytokines data that were previously shown in the literature (as reviewed in [64,65,66] to be related to neuroinflammation and Alzheimer’s disease, including IL-1α, IL-1β, IL-6, TNF-α, IL12-p40, Eotaxin, MIP-1a, and MCP1 (Figure 8B)). For the cytokines such as IL-1α, IL-1β, IL-6, IL-12p40, MIP-1a, and MCP1, the treatment with nanoparticle F reduced their level compared with the treatment with nanoparticles alone, where a minimal effect from the nanoparticles alone was observed when compared to the untreated 3xTg AD mice (Figure 8B). In cytokines such as TNF-α and Eotaxin, the nanoparticles-alone treatment reduced the cytokines level compared to the untreated 3xTg AD mice (Figure 8B). The treatment with nanoparticle F further reduced the cytokines level compared to the nanoparticles-alone treatment, highlighting the synergetic effect of the nanoparticles and F12511 (Figure 8B). We used Volcano plots to illustrate the difference between all 31 cytokine expressions between each treatment group (shown in Appendix A). We note that the effects of the nanoparticles and nanoparticle F on the cytokine expression were cytokine specific. Together, these results suggested that the treatment with nanoparticles alone and nanoparticle F reduced most proinflammatory cytokine expressions in the aging 3xTg AD mice; human Tau and HPTau were previously shown to be highly inflammatory in the brain, and the removal of these aggregated protein species could restore proper brain function by ameliorating neuroinflammation. We suspect that the effect of the nanoparticles and nanoparticle F treatment on inflammatory cytokines could have been in part due to the ability of the nanoparticles and nanoparticle F treatment to reduce the Aβ, human Tau, and HPTau levels.

## 3. Discussion

In this study, we aimed to characterize nanoparticle F treatment in vivo by conducting pharmacokinetic studies by using ACAT enzymatic activity assays and HPLC/MS/MS analyses. We provided evidence that nanoparticle F is able to permeate the blood–brain barrier (BBB). We also showed that by a single intravenous (IV) injection, nanoparticle F at a ~46 mg/kg concentration inhibited ACAT in the brain by ~70% for up to 12 h, and in the periphery for up to 24 h. Importantly, the ACAT activity was restored fully in all tissues examined at 48 h after injection, showing that F12511, a hydrophobic compound, was not accumulating in the tissues that we had examined. Previously, F12511 passed the clinically safety test and did not cause adrenal toxicity in animals [36]. To confirm that nanoparticle F does not cause overt toxicity, we treated mice with a daily IV for 7 days with nanoparticles, with or without F12511. Afterwards, histological examinations on the adrenal glands, liver, and brain tissues were performed, and the results showed that no detectable major morphology changes or tissue damages occurred.

Once nanoparticle F was characterized in the WT mice, the therapeutic efficacy of the AD treatment was investigated. The results showed that the nanoparticle F treatment, as well as the control nanoparticles, provided benefits in the advanced-age 3xTg AD mice. The rather unexpected and exciting finding from this work is that the nanoparticles alone and F12511 may have an additive effect on amyloidopathy and on tauopathy. Based on the Western blot results, nanoparticle F and the vehicle nanoparticles both reduced the full-length mutant hAPP and phosphorylated human tau, and this effect was much stronger with nanoparticle F. Furthermore, only nanoparticle F, but not the nanoparticles alone, reduced Aβ1-42 and the total mutant human tau. The Alzheimer’s pathology drives neuroinflammation through proinflammatory cytokines expression in the brain (reviewed in [64]). Treatment with nanoparticle F significantly reduced classical pro-neuroinflammatory cytokines in 3xTg AD mice brains, such as IL-1β, IL-6, and IL-12p40, as well as other proinflammatory cytokines compared to the nanoparticles-alone treatment. In cytokines such as TNF-α and Eotaxin, the treatment with the nanoparticles and nanoparticle F produced additive effect to further dampen the proinflammatory cytokines level in the 3xTg AD mice brains. Our cytokine analyses results on the 3xTg AD mouse brains further corroborate the effects observed on reducing the full-length mutant hAPP, Aβ1-42, and unphosphorylated and phosphorylated human tau, further reinforce the notion that the nanoparticles alone and F12511 produced additive effects. Determining how the additive effect occurs mechanistically requires further investigation. 

Overall, our results show that the DSPE-PEG_2000_ and PC not only increased the amount of F12511 that was encapsulated into the nanoparticles, but they also produced benefits in the therapeutic treatment. DSPE-PEG and PC have been previously suggested to have some benefit in certain disease model systems. Notably, Brown and colleagues tested the effects of using DSPE-PEG micelles in combination with cyclodextrin as a potential therapy for Niemann–Pick Type C1 (NPC1) disease, an adolescent lysosomal storage disorder [67]. The use of 2-Hydroxy-propyl-b-cyclodextrin (HPβCD) has been used in clinics to treat NPC 1 disease patients, but they required multiple high doses, which resulted in ototoxicity or hearing loss. Utilizing DSPE-PEG not only had an effect itself but worked synergistically with HPβCD in a cell culture [67]. A separate study looking at an AD mouse model utilized liposomes with phosphatidic acid (PA), which is a metabolite of PC. PA was shown to interact with Aβ1-42 [68] and was utilized, along with transferrin and a neuroprotective peptide, to generate liposomes to reduce amyloids in AD mouse models [69]. Yang and colleagues used a trans well migration assay where microglia could migrate to different chambers. The results showed that microglial chemotaxis towards the Aβ-oligomer wells was increased in both the PA liposomes with and without the neuroprotective peptide of interest, suggesting that the PA liposome itself was beneficial. These PA liposomes reduced amyloid; however, to our knowledge, the effect of the liposomes has not been documented with regard to tauopathy. 

One important note is the possibility for nanoparticle F to exhibit a more pronounced effect in mice in the advanced-age range due to the BBB integrity. It has been well established that BBB integrity and function are compromised in certain diseases, including neurodegenerative diseases such as AD. In addition, in aged mice, there is a dysfunction of BBB tight junctions, and this is accompanied by an increase in neuroinflammation [70]. We speculate that the benefits seen in the advanced-age group may be due to a “leaky” BBB that allows more nanoparticle F to enter the brain parenchyma. As mentioned above, F12511 was detectable in the brain by HPLC/MS/MS at a low concentration (~10–20 nM); however, this value was obtained by using WT mice at 2–4 months of age. With a compromised BBB, it is possible that the F12511 concentration in the brain is a lot higher. Further investigations on nanoparticle F pharmacokinetics in aging 16–20-month-old 3xTg AD mice is required to validate this hypothesis. Additionally, future studies are needed to test if modifying the chemical structure of F12511 can increase its potency for ACAT1 inhibition. 

We had previously shown that in the lipopolysaccharide (LPS)-induced acute neuroinflammation mouse model, the genetic KO of ACAT1 in the myeloid cell lineage attenuated neuroinflammation induced by LPS [20]. Mechanistic studies showed that the ACAT1 blockade acted in part by increasing the endocytosis of the receptor TLR4 that mediates the LPS-mediated proinflammatory signaling cascade at the plasma membrane [20]. Acute neuroinflammation and chronic inflammation share many causal factors in common [71,72]. In AD, Aβ, Tau, and hyperphosphorylated Tau are associated with chronic inflammation. We speculate that nanoparticle F produces anti-inflammation in 3xTg AD mice, and this may occur by suppressing these toxic protein species. Additionally, nanoparticle F may also act to alter the fate of TLR4 in microglia or astrocytes. Currently, we do not know how nanoparticles alone act in vitro and in vivo.

Fisher and colleagues showed that in monocyte/macrophages, the acute treatment of apoA1 in vitro and in vivo produce anti-inflammatory responses by promoting a cholesterol efflux at cholesterol-rich lipid rafts, independent of the lipid efflux protein ABCA1 [73]. We suspect that the action of the nanoparticles alone may have acted in similar manner as apoA1: by promoting a cholesterol efflux at the lipid rafts region of the plasma membrane [20,74]. We speculate that blocking ACAT1 may increase the cholesterol content at the lipid raft domain in various membrane organelles, and it maybe through the cholesterol rich lipid raft domain that F12511 and nanoparticles alone act synergistically to benefit AD and other neurodegenerative diseases. Future investigations will be required to dissect the mechanism(s) involved in nanoparticle F suppressing Aβ and/or Tau/hyperphosphorylated Tau and in neuroinflammation. During the preparation of this manuscript, Valencia-Olvera et al. [75] reported that the oral feeding of CI1011 (also named avasimibe) for 60 days ameliorated amyloidopathy and other AD-like pathologies in the 5xFAD mouse model for Alzheimer’s Disease.

Drug delivery into the brain possesses unique challenges due to the presence of the BBB. This problem requires different innovative strategies to achieve the desired payload in the brain [76]. While our current nanoparticle system is sufficient to achieve ACAT inhibition after 4 h of administration, we could further explore active targeting strategies to direct these nanoparticles to the brain to enhance nanoparticle uptake across the BBB. A wide spectrum of different ligands has been investigated, including RVG peptides, cardiolipin, mannose, ApoE, etc., which are potential candidates to combine with our nanoparticle system [76]. Future work will be focused on nanoparticle modification to actively target these nanoparticles into the brain.

## 4. Materials and Methods

### 4.1. Ethical Handling of Animals

The Institutional Animal Care and Use Committee (IACUC) at Dartmouth approved all the mouse experiments under the protocol (#00002020). The genetically ablated *Acat1^−/−^* mice [77] had a C57BL/6 genetic background. The triple transgenic Alzheimer’s disease (3xTg AD) mice [61] with or without *Acat1* [27,28] maintained in the Chang lab had a mixed 129:C57BL/6 genetic background. The mice were subjected to treatment by a tail vein intravenous injection, retro-orbital venous sinus injection or oral gavage depending on the experiment, following the protocol established by the Jackson Laboratory. Depends on the assay, the mice were either administered nanoparticle F (5.8 mg/kg or 46 mg/kg) or nanoparticle K604 (40 mg/kg).

### 4.2. Lipids, ACAT Inhibitors, Solvents, and Chemicals

DSPE-PEG_2000_ was from Laysan Bio, Inc. (mPEG-DSPE, molecular weight 2000, Arab, AL, USA). L-α-Phosphatidylcholine (from egg yolk) was from Sigma-Aldrich (Catalog No. P-2772, St. Louis, MO, USA). F12511 and K604 were custom synthesized by WuXi AppTec in Shanghai, China. Based on the HPLC/MS and NMR profiles, F12511 was 98% in chemical purity and in stereospecificity (F12511 contains an asymmetric center) [51] and K604 was 98% in chemical purity. CP113,818 was a research gift from Pfizer. [1,1-dioctadecyl-3,3,3,3-tetramethylindotricarbocyanine iodide] DiR (catalog #22070) was purchased from AAT Bioquest, Pleasanton, CA, USA.

### 4.3. Nanoparticle Formation

The protocol is described in detail in [48]. Briefly, DSPE-PEG_2000_ dissolved in ethanol (EtOH) and phosphatidylcholine (PC) dissolved in chloroform were combined under a vortex. ACAT inhibitors (F12511 or K604) or a DiR tracer that were dissolved in EtOH were then added to the DSPE-PEG_2000_/PC mixture while under rapid mixing. The final solution contained 30 mM DSPE-PEG_2000_, 0–6 mM PC, and a 1.51–12 mM ACAT inhibitor or 8.3 mM DiR. The final solution was lyophilized overnight (12–16 h) at −40 °C and a 133 × 10^−3^ mBar vacuum setting and stored away from light at −20 ºC. Prior to use, the frozen sample was resuspended in a sterile 1× phosphate-buffered saline (PBS) and bath sonicated in a Branson 2510 sonicator at 4 °C for 2–4 times each round, 20 min per round. The sonicated solution kept in sterile condition was collected in sterile Eppendorf tubes and centrifuged at 12,000 rpm for 5 min to remove the unincorporated materials. The supernatant contained the sterile nanoparticles that were used for treatment directly. In the initial studies, DSPE-PEG_2000_ nanoparticles were employed and loaded with F12511 at varying concentrations. The concentration ranged from a low dose of 1.51 mM (4 mol% applied to the mouse at 5.8 mg F per kg body weight) to a high dose of 12 mM (25 mol% applied to the mouse at 46 mg F per kg body weight). The nanoparticles underwent the same lyophilization procedure. Subsequently, after resolubilization in 1 mL of PBS, the nanoparticles were probe sonicated on ice under sterile conditions by using a Branson probe sonicator. The sonication procedure consisted of two sets of 1 min pulses with a 5 min rest period between sets. The resulting nanoparticles were then used for injection.

### 4.4. Measuring ACAT Activity by Mixed Micelle

This method was described in [49], and its use in mouse tissues was described in [27,28]. WT C57BL/6 mouse tissues (forebrain, cerebellum/brain stem, adrenal glands, and liver) were homogenized by using the Next Advance Bullet Blender homogenizer with stainless steel beads; they were mixed twice at a scale of 8, with each mixing session lasting 3 min, while maintaining a temp at 4 °C. The buffer used contained 2.5% 3-((3-Cholamidopropyl) dimethylammonio)-1-propanesulfonate (CHAPS) and 1 M KCl in 50 mM Tris at pH 7.8. Aliquots of the tissue homogenates were then transferred to prechilled glass tubes that contained the mixed liposomal mixture of 9.3 mM taurocholate, 10.8 mM phosphatidylcholine (egg yolk), and 1.8 mM cholesterol. The tubes were vortexed and kept on ice. The samples were then incubated in a 37 °C shaking water bath with 10 nmole ^3^H-oleoyl CoA/BSA added to start the enzyme reaction for 10 min. The assay was stopped by adding CHCl_3_:MeOH (2:1), vortexed, and centrifuged at 500 rpm for 10 min. The top phase was removed, and the bottom phase was blow dried by N2, and 100 µL of ethyl acetate was added to each tube, vortexed, and spotted on a thin-layer chromatography (TLC) plate (Miles Scientific Silica gel HL, Catalog No. P46911). The TLC solvent system used was petroleum ether:ethyl ether:acetic acid (90:10:1). The cholesteryl ester band with an Rf value at 0.9 was visualized by iodine staining, scraped from the plate, and measured by a scintillation counter for radioactivity. 

### 4.5. Histology

WT C57BL/6 mice were either left untreated, treated with control DSPE-PEG2000/PC nanoparticles, or nanoparticles that contained F12511 (Nanoparticle F). The control and nanoparticle-F-treated mice received 7 daily IV injections and were sacrificed 48 h after the last injection. The mice were anesthetized with Avertin (2% in PBS) and, after confirming by a lack of a toe-pinch reflex, were slowly perfused with 10 mL of 4% sucrose in PBS followed by 10 mL of 4% paraformaldehyde (PFA) in a 4% sucrose solution in PBS. The tissues were collected and kept in PFA at 4 °C on a sample mini rotator overnight before switching the solution to 4% sucrose in the PBS. The tissues were then collected into histology cassettes, embedded into paraffin blocks, and sectioned onto slides with a hematoxylin and eosin (H&E) stain. 

### 4.6. HPLC/MS/MS

F12511 was quantified in mouse tissues via LC-MS/MS and CP113,818 (a research gift from Pfizer), and a closely related ACAT inhibitor was used as the internal standard. Both compounds were dissolved in DMSO at 10 mg/mL and stored at −40 °C. Subsequent working dilutions were made in fresh acetonitrile (ACN) daily. Calibrators and quality control measures were used in the appropriate matrix: C57BL/6 plasma (anticoagulant: K3-EDTA, Innovative Research), brain homogenate, liver homogenate, and adrenal gland homogenate. The tissues were homogenized at 0.1 g/mL in diH_2_O by using stainless steel beads and a Next Advance Bullet Blender. All the samples (50 µL) were protein precipitated, with 150 µL of 10 ng/mL CP113,818 in ACN acting as an internal standard and added via vortex for 1 min and centrifugation for 5 min at 15,000 rpm. In total, 150 µL of supernatant was transferred to amber autosampler vials, and 10 µL was injected on the LC-MS/MS system. HPLC separation was achieved with isocratic conditions of 5% diH_2_O, 95% methanol, and 0.1% formic acid over 2.5 min at a flow rate of 1.5 mL/min on a Phenomenex Luna C18 100 × 4.6 mm and 3 micron column fitted with a 10 × 4 mm C18 guard at 40 °C. A TSQ Vantage mass spectrometer was operated in positive ion mode with a collision pressure of 1.3 mTorr to measure F12511 (470.242→268.120 *m*/*z*) and CP113,818 (471.177→201.040 *m*/*z*) with collision energies of 16 and 21, respectively, and S-Lens values of 139 and 187, respectively. The heated ESI source was operated with a spray voltage of 4500 V, vaporizer temperature at 500 °C, capillary temperature at 250 °C, and sheath and auxiliary gases at 30 and 15 arbitrary units, respectively. The quantitative range for F12511 was 0.3–1000 ng/mL for the samples from the liver, adrenal glands, brain, and whole blood, and 0.5–1000 ng/mL for the samples from the plasma.

### 4.7. Preparation of Brain Homogenates and Western Blot Analysis

Half of the forebrains from the male and female 3xTg AD mice at the indicated age ranges were homogenized at 4 °C in the Bullet Blender with stainless steel beads in the sucrose buffer with protease inhibitors, as described in previously published studies [78]. The samples were centrifuged at 4 °C for 5 min at 12,000 rpm to collect any precipitate. The homogenates were then collected for protein analysis by a Lowry assay and divided into prechilled Eppendorf tubes and stored at −80 °C until ready for analysis. For the Western blot analysis, loading dye was added such that the final sample loading concentration contained at least 10% sodium dodecyl sulfate (SDS). Western blots for detecting human mutant full-length human mutant APP were described in [27], and Western blots to detect human tau and hyperphosphorylated tau were described in [32]. The blots were developed by using Odyssey DLx LI-COR and analyzed by using Image Studio Software Version 1.0.31 by LI-COR. 

### 4.8. Antibodies

The mouse antihuman amyloid beta1-42 (6E10), mouse antihuman tau (HT7) and mouse antihuman hyperphosphorylated tau (PHF-tau (AT8)), and anti-beta-actin (beta-actin as the loading control) were from BioLegend (San Diego, CA, USA), Thermo Fisher (Waltham, MA, USA), and Sigma-Aldrich (St. Louis, MO, USA), respectively. 

### 4.9. ELISA for Monitoring Human Amyloid Beta_1–42_ (Aβ1-42)

The tissue homogenates were prepared as described above. The homogenates underwent a formic acid extraction following previously published protocol [79]. The samples were then measured by using a human Aβ1-42 enzyme-linked immunoassay (ELISA) kit (Catalog No. KHB3441, Invitrogen by Thermo Fisher Scientific).

### 4.10. Monitor Biodisitrbution of Nanoparticle with DiR

This method was adapted from [57] with modifications. Briefly, 3-month-old WT female mice were IV injected with DiR nanoparticles. Four hours later, the mice were sacrificed and perfused with ice cold 1xPBS. Their brains were then collected and separated into forebrain and cerebellum regions. Each brain region was then homogenized in 5 mM EDTA and 5% Acetone at 4 °C in the Bullet Blender with stainless steel beads and loaded into microhematocrit nonheparinized capillary tubes (Fisher Scientific, Hampton, NH, USA) for the fluorescence quantification. Images were captured on the Pearl 500 imaging system (LICOR Biosciences, Lincoln, NE, USA) and quantified by using Fiji-ImageJ Software, version 2.1.0/1.53c.

### 4.11. Brain Luminex Cytokine Analysis

Cytokines from the mouse brain homogenates prepared in Section 4.7 were measured by using Millipore mouse cytokine multiplex kits (EMD Millipore Corporation, Billerica, MA, USA). Calibration curves from the recombinant cytokine standards were prepared by following threefold dilution steps in the same matrix as the samples. High- and low-quality control samples with a known concentration provided by the manufacturer were utilized to validate the calculation of the standard curve. The standards and quality-control samples were measured in triplicate, while the samples were measured once, and blank values were subtracted from all the readings to ensure an accurate measurement. All the assays were carried out directly in a 96-well filtration plate (Millipore, Billerica, MA, USA) at room temperature and protected from light. Briefly, each well was prewet with 100 µL of PBS containing 1% BSA. Then, the beads, along with a standard, sample, quality control samples, or blanks were added in a final volume of 100 µL. The plate was incubated at room temperature for 30 min with continuous shaking. The beads were washed three times with 100 µL of PBS containing 1% BSA and 0.05% Tween 20. A cocktail of biotinylated antibodies (50 µL/well) was added to the beads for a further 30 min of incubation with continuous shaking. The beads were washed three times and then streptavidin-PE was added for 10 min. The beads were again washed three times and resuspended in 125 µL of PBS containing 1% BSA and 0.05% Tween 20. The fluorescence intensity of the beads was measured in using the Bio-Plex array reader 200 from BioRad. Bio-Plex Manager software Version 6.2 with five-parametric-curve fitting was used for the data analysis.

### 4.12. Oral Gavage to Mice

The gavage tube was filled with the volume equivalent to 10 mL/kg of the mouse weight, with either the nanoparticles alone or nanoparticle F at the dose of 46 mg F12511/kg. The mice were restrained by tightly scruffing them with one hand. The gavage tube was then placed in the diastema of their mouth and then advanced along the upper palate until the esophagus was reached. The plunger was slowly and steadily depressed to administer the solution into the stomach. After the injection, the gavage needle was removed and the mouse was placed in a clean and comfortable environment to recover and to be monitored.

## 5. Patents

Dartmouth College has filed a U.S. Provisional Application entitled “Method for Attenuating Neuroinflammation, Amyloidopathy and Tauopathy”.

## Figures and Tables

**Figure 1 ijms-24-11013-f001:**
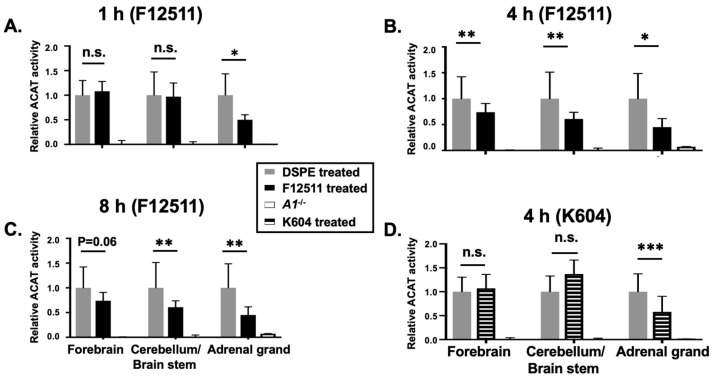
IV injection of nanoparticle F reduces ACAT activities in both adrenal glands and brain, while nanoparticle K604 only reduces ACAT activity in adrenal glands but not in brain. WT mice were IV injected with either nanoparticle F with F12511 at low concentration (30 mM DSPE-PEG2000 with 5.8 mg F per kg mouse bodyweight) or with empty nanoparticle at zero time and sacrificed after (**A**) 1 h, (**B**) 4 h, or (**C**) 8 h. (**D**) WT mice were IV injected with nanoparticle K604, with K604 at high concentration (30 mM DSPE-PEG2000 with 24 mg K604 per kg) or with empty nanoparticle at zero time and sacrificed after 4 h. *A1^−/−^* refers to *Acat1^−/−^* mouse as a negative control. Relative ACAT activity was determined by using the ACAT enzyme assay as described in the methods. WT mice per group, at age 4–5 months with gender matched. N = 1 for *A1^−/−^* mouse. Two-way ANOVA was conducted for statistics. *p* < 0.001 ***; *p* < 0.01 **; *p* < 0.05 *; n.s., not significant.

**Figure 2 ijms-24-11013-f002:**
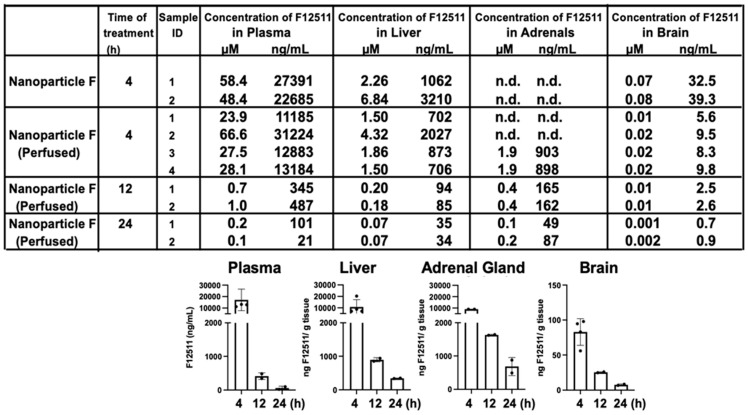
HPLC/MS/MS analyses of F12511. HPLC/MS/MS results (in (**top**) panel) shown F12511 content in the WT mouse plasma, liver, adrenal glands, and brain after 4, 12, and 24 h post IV delivery of nanoparticle F (at 46 mg F per kg mouse body weight. Most of the mice were perfused with PBS for 15 min to avoid blood contamination of tissues before sample collections. For each mouse tissue measured, each point represents the average of 3 replicates, with 3 HPLC column injections per replicate. DSPE-PEG_2000_/PC-treated mice were also measured by HPLC/MS/MS. As expected, values for F12511 were under the detectable limit. Results from the table on (**top**) were replotted in graphs shown at (**bottom**) by using results from perfused tissue.

**Figure 3 ijms-24-11013-f003:**
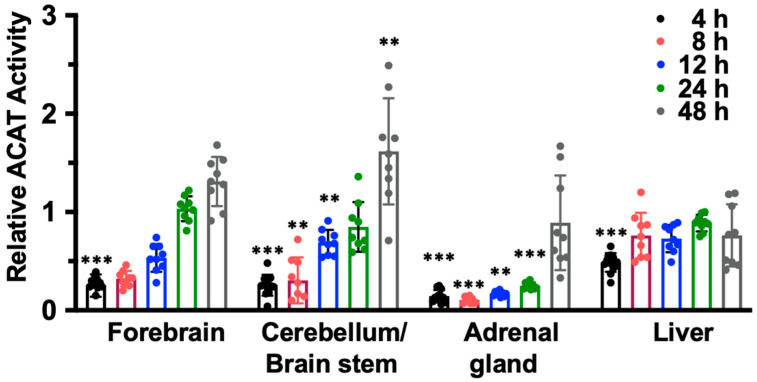
The ACAT activity in forebrain, cerebellum, liver, and adrenals gradually return to normal 24 to 48 h after a single IV of nanoparticle F. Adult WT mice were given single IV of nanoparticle F at high dose (~46 mg/kg). At various time points indicated, mice were perfused with PBS. The forebrain, cerebellum/brain stem, adrenal gland, and liver were isolated and homogenized to measure ACAT enzyme activity. One-way ANOVA was conducted to determine statistics. *p* < 0.001 ***, *p* < 0.01 **.

**Figure 4 ijms-24-11013-f004:**
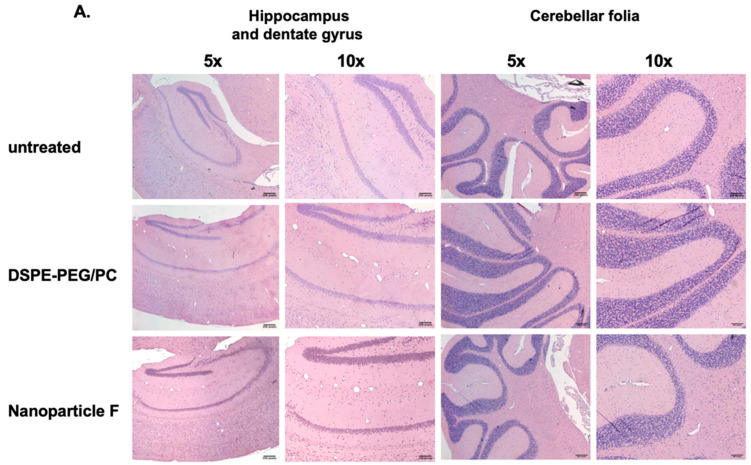
Treating normal mice with nanoparticle F or nanoparticles alone produces no overt morphological alterations in central nervous and peripheral tissues. Representative images from WT mice (n = 3–4/group) either untreated or treated with IV injections once per day with DSPE-PEG_2000_/PC or with nanoparticle F for 7 days. Tissues were collected 48 h after the last injection. (**A**) Images show hippocampus and dentate gyrus regions (left panels). Images on the right panels show cerebellar folia region. (**B**) Images show adrenal glands (left panels), and livers (right panels). For adrenal gland images, symbols used: m = medulla; ZR = zona reticularis; ZF = zona fasciculata; ZG = zona glomerulosa. Scale bars: 200 pixels: ~109 µm for 10× enlargement and ~54 µm for 20× enlargement. Scale bar: 200 pixels.

**Figure 5 ijms-24-11013-f005:**
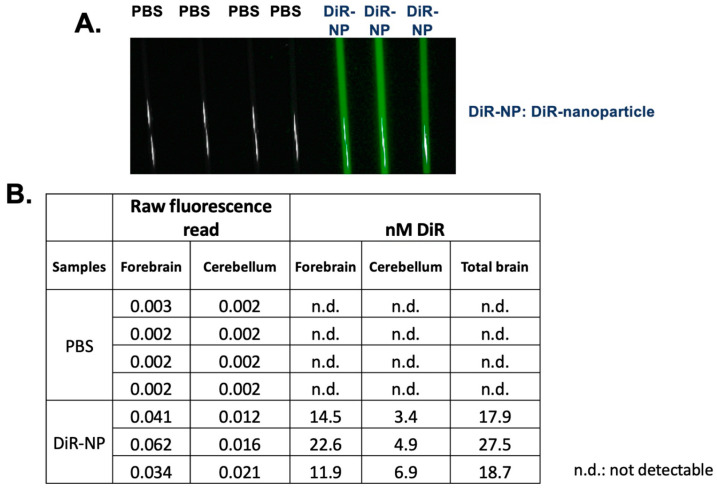
Biodistribution of nanoparticles. 3-month-old sex-matched WT mice were injected with 8.3 mM of 1,1′-Dioctadecyl-3,3,3′,3′-Tetramethylindotricarbocyanine Iodide (DIR) encapsulated in DSPE-PEG/PC nanoparticles (NP). Mice were sacrificed after 4 h and perfused with 20 mL of cold 1xPBS. Brains were collected and separated into 2 hemispheres. The right hemisphere was homogenized in lysis buffer and loaded into capillary tubes for imaging capture on the Pearl 500. (**A**) Representative of mouse brain homogenates in capillary tubes. (**B**) Quantitative analysis of DiR in mouse brain homogenate. Raw fluorescence read was applied to the generated standard curve to calculate the amount of DỉR in the brain homogenate of injected mice. All images were analyzed in Fiji. N = 3–4 mice/treatment group. Scale bar = 2500 μm. n.d.: not detectable.

**Figure 6 ijms-24-11013-f006:**
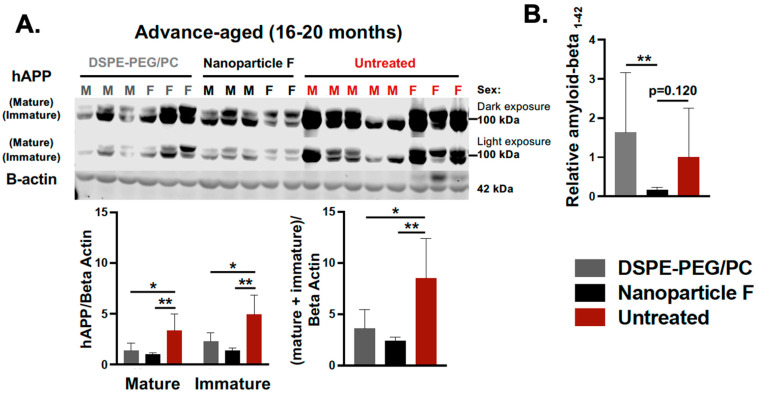
IV injections of nanoparticle F reduce mutant hAPP and reduce total acid extractable Aβ1-42 in advance-aged 3xTg AD mice. Male (M) and female (F) 3xTg AD mice at 16–20 months of age were either untreated or treated with DSPE-PEG_2000_/PC (vehicle) or with nanoparticle F (F12511 at 46 mg/kg), as indicated with daily IV injections once daily for 2 weeks. Mouse brain homogenates were prepared according to procedure described in PMID: 15980612 and PMID: 20133765. (**A**) Western blots were used to monitor the immature and mature forms of hAPP (two adjacent bands at 105-kDa and 115-kDa) by using mouse monoclonal antibody anti-6E10 (recognizing amyloid Aβ; 1:5000 from Covance). Results are shown as light and dark exposures. Two-way ANOVA was conducted to analyze the signals for mature and immature bands individually. One-way ANOVA was used to analyze total signals comprising both mature and immature bands. Mouse anti-beta-actin antibodies were used as loading controls. (**B**) ELISA assay used on mouse monoclonal antibody anti-6E10 to monitor total formic acid extractable Aβ1-42 levels in brain homogenates. Procedure for Aβ1-42 extraction was described in PMID: 15980612 and PMID: 20133765. ELISA plates were from Invitrogen. Each sample was measured in quadruplicate. Results are presented as Aβ1-42 signal intensity from mice treated with DSPE-PEG_2000_/PC (nanoparticle/vehicle) or with nanoparticle F relative to that of untreated control using one-way ANOVA analysis. N.S. not significant; *p* < 0.01 **, *p* < 0.05 *.

**Figure 7 ijms-24-11013-f007:**
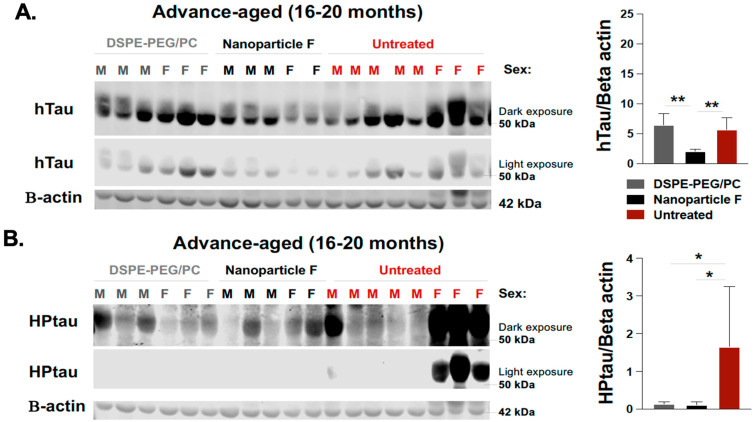
Nanoparticle F reduces total unphosphorylated mutant human tau (hTau) and hyperphosphorylated human tau (HPTau) while nanoparticles alone reduce HPTau but do not reduce htau in advance-aged 3xTg AD mice. The same brain homogenate samples described in Figure 6 were used for Western blot analyses to monitor hTau (**A**) and HPTau (**B**), according to procedure described in PMID: 25930235. Mouse anti-HT7 used for total unphosphorylated human tau (~50 kDa) and mouse anti-AT8 used for hyperphosphorylated human tau (~55–60 kDa) were from Thermo Fisher Scientific, and mouse anti-beta-actin (42 kDa) antibodies served as loading control. Dark and light exposure of each blot is shown. One-way ANOVA was conducted to determine statistics. N.S. not significant; *p* < 0.01 **, *p* < 0.05 *.

**Figure 8 ijms-24-11013-f008:**
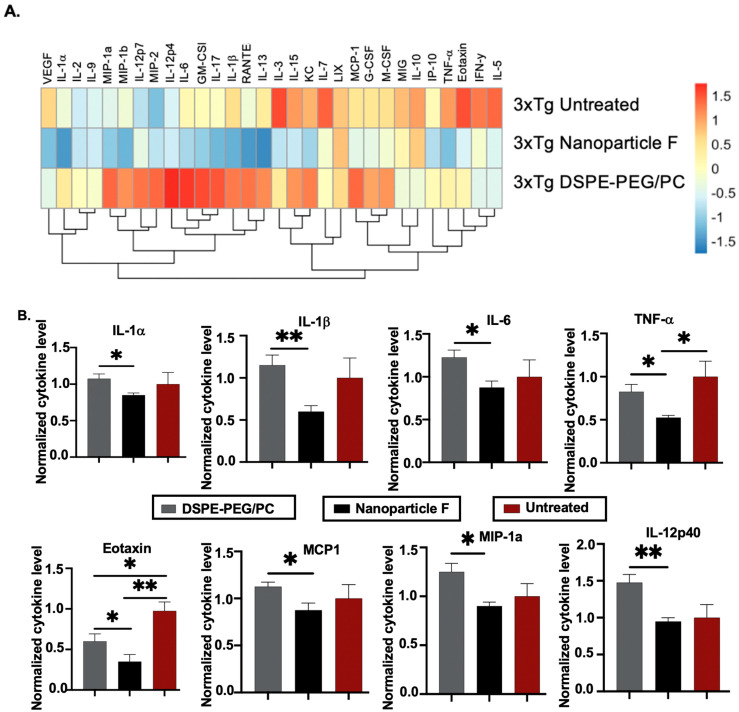
DSPE-PEG/PC and nanoparticle F alter proinflammatory cytokines profile in 3xTg AD mice (16–20 month after 2 weeks of daily IV/RO injections). Mice were perfused with 20 mL of cold 1xPBS and their brains were collected and homogenized. Forebrain homogenates were analyzed by using MILLIPLEX MAP Mouse Cytokine/Chemokine Magnetic Bead Panel (32 plex), and data were normalized by using total protein content determined by Lowry assay. (**A**) Heatmap visualizing average cytokines readings from several biological replicates of each treatment group with a Z-score transformation. (**B**) Alzheimer’s-related cytokines were plotted individually with 3xTg AD mice forebrain cytokines content. For B, values obtained from each individual cytokine from PBS injected animal is normalized to 1. Unpaired student *t*-test (two tailed) were performed; *p* < 0.01 **, *p* < 0.05 *. N = 4 mice/group.

## Data Availability

Not applicable.

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
