# Peer review of "Stealth Liposomes Encapsulating a Potent ACAT1/SOAT1 Inhibitor F12511: Pharmacokinetic, Biodistribution, and Toxicity Studies in Wild-Type Mice and Efficacy Studies in Triple Transgenic Alzheimer’s Disease Mice"

_ijms, 2023, doi:10.3390/ijms241311013_

Round 1

Reviewer 1 Report

The manuscript, “Stealth liposomes encapsulating a potent ACAT1/SOAT1 inhibitor F12511: pharmacokinetic, biodistribution and toxicity studies in wild-type mice, and efficacy studies in triple transgenic Alzheimer Disease mice.” determines kinetics, biodistribution, and toxicity in WT mice as well as possible reduced pathology involved with AD.  The authors use an ACAT1 inh F12511 encapsulated in a DSPE-PEG2000/PC protected nanoparticle.  This manuscript is interesting as the nanoparticle F could increase therapeutic efficacy of F12511.  However, there are several major issues with the manuscript as it stands. 

The authors have left out several figures that prevent the proper analysis of this manuscript.  Truthfully, this issue makes me question the validity of the manuscript as a whole.  The authors omitted Fig. 3 and replaced it with figure 4A.  Figure 4C is omitted.  Finally, Fig. 7 is omitted.  These omissions are unacceptable.

In the introduction, the authors suggest BBB permeability of F12511 is unknown (line 122).  Interestingly, the authors did not measure the BBB penetrance of the F12511 molecule. The authors then suggest in the discussion (Line 418), “Future studies are needed to test if modifying the chemical structure of F12511 to increase its BBB permeability.” but the question is why?  I believe the idea of using a nanoparticle is to reduce drug metabolism but also circumvent the BBB by passive endocytosis.  Would modifying F12511 affect brain parenchymal uptake in this nanoparticle system?  Please expand the discussion on this topic. 

Importantly, the authors need to measure F12511 in perfused mouse brain without nanoparticles.  We do not know if F12511 passes BBB or have an appropriate control of F12511 treatment concentration in perfused mouse brain.

Why did the authors choose not to measure ACAT activity or F12511 concentration in 3xTg AD mice?

Minor criticisms

Why was t-test used for fig 8b?

Figure 6a, first band is cropped.

Wording line 426 and 428

Please stay consistent with use of nanoparticle F vs. F12511 vs. nanoparticle F12511.

Figure 1. legend says DEPC, is this correct?

Author Response

Reviewer #1- Comments and Suggestions for Authors 

  1. The manuscript, “Stealth liposomes encapsulating a potent ACAT1/SOAT1 inhibitor F12511: pharmacokinetic, biodistribution and toxicity studies in wild-type mice, and efficacy studies in triple transgenic Alzheimer Disease mice.” determines kinetics, biodistribution, and toxicity in WT mice as well as possible reduced pathology involved with AD.  The authors use an ACAT1 inh F12511 encapsulated in a DSPE-PEG2000/PC protected nanoparticle.  This manuscript is interesting as the nanoparticle F could increase therapeutic efficacy of F12511.

 We appreciate the reviewer's positive feedback on our manuscript.

  1. However, there are several major issues with the manuscript as it stands.   

The authors have left out several figures that prevent the proper analysis of this manuscript.  Truthfully, this issue makes me question the validity of the manuscript as a whole.  The authors omitted Fig. 3 and replaced it with figure 4A.  Figure 4C is omitted.  Finally, Fig. 7 is omitted.  These omissions are unacceptable. 

Thank you for bringing this to our attention. We have recently discovered that two figures, Figure 3 and Figure 7, were inadvertently deleted in the version of the manuscript sent to the reviewers. We believe this deletion occurred during the manuscript processing stage, and we will verify with the editor, Dr. Tamara Ugarkovic, who oversees the review process, to confirm that it was an error made by the journal's processing system.

In the revised manuscript, we have included the missing Figure 3 and Figure 7 to ensure the completeness of the study.

To provide full transparency, we would like to acknowledge that we had previously submitted the same manuscript to the preprint server BioRxiv, where it is publicly available. The version on the preprint server includes all the figures that were missing in the initial submission to IJMS.

Regarding Figure 4C, we appreciate your observation. It should actually refer to Figure 4B, which presents the adrenal gland histology data. We have rectified this mistake in the revised manuscript.

Thank you once again for your valuable feedback, and we apologize for any confusion caused by these errors.

  1. In the introduction, the authors suggest BBB permeability of F12511 is unknown (line 122). Interestingly, the authors did not measure the BBB penetrance of the F12511 molecule. The authors then suggest in the discussion (Line 418), “Future studies are needed to test if modifying the chemical structure of F12511 to increase its BBB permeability.” but the question is why? I believe the idea of using a nanoparticle is to reduce drug metabolism but also circumvent the BBB by passive endocytosis.

Thank you for your comment. We have revised the sentence as follows:

"Future studies are needed to test if modifying the chemical structure of F12511 can increase its potency for ACAT inhibition." (now line 788-789).

  1. Would modifying F12511 affect brain parenchymal uptake in this nanoparticle system? Please expand the discussion on this topic.

Thank you for your question. We are actively engaged in pursuing further investigations in this area.

  1. Importantly, the authors need to measure F12511 in perfused mouse brain without nanoparticles.  We do not know if F12511 passes BBB or have an appropriate control of F12511 treatment concentration in perfused mouse brain. 

Thank you for your question. We recognize the solubility challenges associated with F12511, which is a highly hydrophobic compound, and agree that it cannot be easily delivered alone into mice. In our manuscript (line 214-216), we have mentioned the need for a vehicle, such as cyclodextrin, to enhance solubility in aqueous formulations, as referenced (Reference 51,52).

Since direct injection of F12511 without a carrier is not feasible as it would precipitate in the blood. However, we conducted an experiment to indirectly address your question.

We utilized oral gavage delivery of nanoparticle F12511 (46mg/kg) in mice, once-daily for 4 days. 6th hours after the last administration, we measured ACAT activity in the adrenal glands, liver, and brain of the mice. Using this delivery method, the nanoparticle is expected to degrade in the intestine (reference 53), releasing F12511 into the bloodstream. The results demonstrated that at the 6-hour time point, this delivery method partially inhibited ACAT in the adrenal glands and liver but did not inhibit ACAT activity in the brain. This suggests that without the nanoparticle, F12511 is not readily permeable to the brain. These findings are now included in Figure S1, and the description of this result has been updated accordingly from line 211-225.

  1. Why did the authors choose not to measure ACAT activity or F12511 concentration in 3xTg AD mice? 

Thank you for your question. In our study, we chose to treat 3xTg AD mice "chronically" for a duration of 2-weeks in order to assess the efficacy of the treatment in these valuable aged mice. As for the wild-type (C57BL/6) mice, we used them to measure ACAT activity and monitor F12511 concentration, primarily because they are more readily available for experimental purposes.

Moving forward, we recognize the importance of investigating the concentration of F12511 and ACAT activity specifically in aging 3xTg AD mice. This will allow us to explore whether the compromised blood-brain barrier (BBB) in aging 3xTg AD mice facilitates increased entry of nanoparticle F into the brain. We have now addressed this point in the Discussion section of our manuscript (Line 790-793).

Minor criticisms:

(1) Why was t-test used for fig 8b?

Thank you for your question. To ensure the appropriate statistical analysis, we sought guidance from the Dartmouth Data Analytics Core Facility. Based on their recommendation, we opted to use a t-test for our analysis. This choice was made because we were comparing two groups of samples at a time, rather than evaluating all three groups together (i.e., PBS vs DSPE/PC, DSPE/PC vs nanoparticle F, PBS vs nanoparticle F). By using the t-test, we were able to effectively assess the statistical significance of the observed differences between the specific groups being compared.

(2) Figure 6a, first band is cropped

Thank you, we have taken immediate action to correct the mistake.

(3) Wording line 426 and 428. 

Thank you, we have corrected this. 

(4) Please stay consistent with use of nanoparticle F vs. F12511 vs. nanoparticle F12511. 

Thank you, we have made the necessary updates to ensure consistency throughout the manuscript.

(5) Figure 1. legend says DEPC, is this correct? 

Thank you, we have correct this mistake to DSPE

Reviewer 2 Report

First of all, congratulations on the good work done. It is perfectly structured and very easy to understand. I just wanted to make some comments and if possible solve some questions. Thanks in advance.

1. In the materials and methods section, section 4.3 for the production of nanoparticles, when the solution is lyophilized, which lyophilization cycle was established? Time, temperature, vacuum conditions?

2. In that same section, it is also mentioned that a sonication is carried out (line 466), for how long and what is the purpose of sonication?

3. Line 467 specifies that the sonicated solution was kept under sterile conditions, what measures were used or how was the sterilization of the solution carried out?

4. In the results section, on line 134, include a period instead of a comma.

5. I recommend improving the sharpness of the table included in figure 2.

6. In the text, in line 210, comments are made on figures 4A and 4B, however, only one B appears in the figure, although the figure caption does distinguish both. Lines 212 and 216 also refer to Figure 4C which is not located.

7. And in figure 3, an A appears. It is as if in a previous version both images were part of the same figure and now they are called figures 3 and 4. Please try to clarify.

8. On line 304 the caption of figure 7 appears, but there is no figure included in that part.

Author Response

Reviewer #2 - Comments and Suggestions for Authors 

First of all, congratulations on the good work done. It is perfectly structured and very easy to understand. I just wanted to make some comments and if possible solve some questions. Thanks in advance. 

We appreciate the reviewer's positive and encouraging feedback on our work. The support is valuable to us, and we will continue to strive for excellence in our research. 

  1. In the materials and methods section, section 4.3 for the production of nanoparticles, when the solution is lyophilized, which lyophilization cycle was established? Time, temperature, vacuum conditions? 

Thank you for your question. Temperature -40 oC, time: overnight (12-16 hours), vacuum condition: 133 x 10-3 mBar. We have now included this in the Method section for clarification. (Line 901-902).

  1. In that same section, it is also mentioned that a sonication is carried out (line 466), for how long and what is the purpose of sonication? 

Thank you for your question. We appreciate your feedback and have made the following additions to the method section to provide more clarity.

“The sonication method was performed at 4 °C, using the Branson 2510 bath sonicator model. The process involved sonication for 2-4 cycles, with each cycle lasting 20 min. The purpose of sonication was to disrupt the large multi-lamellar vesicles and promote the formation of smaller vesicles. Additionally, for more detailed information on our nanoparticle formation process, we refer readers to our previously published method paper (De La Torre, et al., 2022, Journal of Neuroscience Methods).” (Line 903-905)

  1. Line 467 specifies that the sonicated solution was kept under sterile conditions, what measures were used or how was the sterilization of the solution carried out? 

Thank you for your question. We appreciate your feedback and have made the following additions to the method section to provide more clarity.

“Before use, all tools were carefully cleaned and wiped with 70% ethanol. The Eppendorf tubes used for these experiments were autoclaved prior to use to maintain sterility. Additionally, all reagents were dissolved in 100% ethanol, dried, and subsequently lyophilized. To facilitate the sonication process, the capped Eppendorf tubes were placed in a bath sonicator.”

  1. In the results section, on line 134, include a period instead of a comma. 

Thank you, we have corrected this. 

  1. I recommend improving the sharpness of the table included in figure 2. 

Thank you, we have made the correction. 

  1. In the text, in line 210, comments are made on figures 4A and 4B, however, only one B appears in the figure, although the figure caption does distinguish both. Lines 212 and 216 also refer to Figure 4C which is not located. 
  2. And in figure 3, an A appears. It is as if in a previous version both images were part of the same figure and now they are called figures 3 and 4. Please try to clarify. 
  3. On line 304 the caption of figure 7 appears, but there is no figure included in that part. 

Thank you for bringing this to our attention. We have recently discovered that two figures, Figure 3 and Figure 7, were inadvertently deleted in the version of the manuscript sent to the reviewers. We believe this deletion occurred during the manuscript processing stage, and we will verify with the editor, Dr. Tamara Ugarkovic, who oversees the review process, to confirm that it was an error made by the journal's processing system.

In the revised manuscript, we have included the missing Figure 3 and Figure 7 to ensure the completeness of the study.

Regarding Figure 4C, we appreciate your observation. It should actually refer to Figure 4B, which presents the adrenal gland histology data. We have rectified this mistake in the revised manuscript.

Thank you once again for your valuable feedback, and we apologize for any confusion caused by these errors.

Reviewer 3 Report

Although scientifically sounds like this manuscript may be interesting, I cannot meaningfully review the entire manuscript due to low presentation quality. Here is some of my comments- An abstract is missing the connection between cholesterol and AD. In addition, the abstract is not reading cholesterol and atherosclerosis are related. Please address this. I would suggest removing atherosclerosis from there and keep the focus on AD. Mention full name for F12511 Line 92, I suggest using the full chemical names for these compounds, at least in the first citation. Nanoliposomes and nanoparticle have been used interchangeably throughout the manuscript. It is confusing. Please address this. Fig 1. What is DEPC in the legend? These figures with the given legends and captions are unreadable to me.

Author Response

Reviewer #3 - Comments and Suggestions for Authors 

Although scientifically sounds like this manuscript may be interesting,

We appreciate the reviewer's positive and encouraging feedback.

I cannot meaningfully review the entire manuscript due to low presentation quality.

We have now enlarged all figures & ensure high resolution for all figures.

Here is some of my comments-

An abstract is missing the connection between cholesterol and AD. In addition, the abstract is not reading cholesterol and atherosclerosis are related. Please address this. I would suggest removing atherosclerosis from there and keep the focus on AD.

Thank you for bringing this to our attention. We have now removed atherosclerosis from the abstract & added the following statement:

In Alzheimer’s Disease (AD), CEs accumulate in vulnerable brain regions. Therefore, ACATs may be promising targets for treating AD.” (Line 20-22).

The goal of adding this statement is to establish the link between cholesterol/CEs metabolism & targeting ACATs to AD.

Mention full name for F12511 Line 92, I suggest using the full chemical names for these compounds, at least in the first citation.

Thank you for your comment & attention to details. We have now included full chemical names for all three ACAT inhibitors when they are first appeared in the manuscript.

Nanoliposomes and nanoparticle have been used interchangeably throughout the manuscript. It is confusing.

Thank you for your comment. We have now made all the necessary adjustments to stay consistent. All “nanoliposomes” term are now changed to “nanoparticles”.

Please address this. Fig 1. What is DEPC in the legend? These figures with the given legends and captions are unreadable to me.

Thank you for raising the question. We have corrected DEPC in the legend to DSPE, which stands for DSPE-PEG2000, one of the main components in our nanoliposome/nanoparticle formulation. Additionally, as mentioned above, we have enlarged all figures & ensure high resolution for all figures.

Reviewer 4 Report

The manuscript entitled “Stealth liposomes encapsulating a potent ACAT1/SOAT1 inhibitor F12511: pharmacokinetic, biodistribution and toxicity studies in wild-type mice, and efficacy studies in triple transgenic Alzheimer Disease mice” is potentially interesting in the field of novel therapies for the treatment of atherosclerosis and Alzheimer disease (AD).

Unfortunately the English is so poor that it is very difficult, impossible to understand the scientific value of the study, starting from the abstract, the introduction and all the rest of the sections are poorly understood.

In my opinion, the manuscript should be edited by a native English and authors should do the effort to much improve the manuscript, also putting in order tables and figures included in the manuscript.

Unfortunately the English is so poor that it is very difficult, impossible to understand the scientific value of the study, starting from the abstract, the introduction and all the rest of the sections are poorly understood.

Author Response

Reviewer #4 - Comments and Suggestions for Authors 

The manuscript entitled “Stealth liposomes encapsulating a potent ACAT1/SOAT1 inhibitor F12511: pharmacokinetic, biodistribution and toxicity studies in wild-type mice, and efficacy studies in triple transgenic Alzheimer Disease mice” is potentially interesting in the field of novel therapies for the treatment of atherosclerosis and Alzheimer disease (AD).

Thank you for your positive comment.

Unfortunately the English is so poor that it is very difficult, impossible to understand the scientific value of the study, starting from the abstract, the introduction and all the rest of the sections are poorly understood.

In my opinion, the manuscript should be edited by a native English and authors should do the effort to much improve the manuscript, also putting in order tables and figures included in the manuscript.

Thank you for your comment & feedbacks. We strive to incorporate all reviewers’ feedbacks to enhance our writing quality.

Since other reviewers did not raise major concern regarding the quality of English language in our manuscript, we will ask the editor to make the final decision on whether additional English revision is required for the current version of our manuscript.

Reviewer 5 Report

In the work by Torre et al, pegylated liposomes were produced for the encapsulation of an ACAT1 inhibitor used for anti-atherosclerosis, that could be promising for Alzheimer’s disease (AD) therapy. The theme is interesting and timely, since finding new therapy strategies for AD is urgent.

The developed nanoformulations were produced and physiochemically characterized in a previous published study and in this work their biodistribution, safety and efficacy were evaluated in mice.

It is a interesting work, but one of its main limitations, is the lack of clarity regarding the goals and the methodologies. For starters, regarding the purpose of encapsulating the K604 inhibitor. The NPs containing K604 were used in this work? For what purpose? The encapsulation of K604 inhibitor is not mentioned in the abstract. Then, these NPs are mentioned in the results (only in the first part), but there is no mention of treating the animals with those in the methods.

Also, in lines 474-489: the authors described an assay using mouse tissues. However, it is not clear from which animals were the tissues extracted. It is from the mice used in this work? And, which tissues were used?

Furthermore, drug delivery to the brain is very challenging and hinders the success of many therapeutic drugs. However, no active targeting strategies were explored by the authors.

Author Response

Reviewer #5 - Comments and Suggestions for Authors 

In the work by Torre et al, pegylated liposomes were produced for the encapsulation of an ACAT1 inhibitor used for anti-atherosclerosis, that could be promising for Alzheimer’s disease (AD) therapy. The theme is interesting and timely, since finding new therapy strategies for AD is urgent. 

We appreciate the reviewer's positive and encouraging feedback.

The developed nanoformulations were produced and physiochemically characterized in a previous published study and in this work their biodistribution, safety and efficacy were evaluated in mice. 

It is a interesting work, but one of its main limitations, is the lack of clarity regarding the goals and the methodologies.  

For starters, regarding the purpose of encapsulating the K604 inhibitor. The NPs containing K604 were used in this work? For what purpose? The encapsulation of K604 inhibitor is not mentioned in the abstract.  

Thank you for your comment. We appreciate your feedback and have made the necessary revisions to clarify our goals and rationale. The following statements have been added to address your concerns:

  1. Line 160: “In order to determine an optimal formulation to inhibit ACAT activity in the brain, we encapsulated F12511 or K604 as part of a DSPE-PEG2000-based nanoparticle.”
  2. Line 185-188: “Overall, our results clearly demonstrate that F12511, when delivered in the nanoparticle system, exhibits superior inhibitory effects on ACAT in the brain compared to K604. Based on these findings, we have decided to shift the focus of our study towards investigating the pharmacokinetics, biodistribution, and efficacy of nanoparticle F12511 specifically in the brain. This strategic shift will enable us to gain a deeper understanding of the potential of nanoparticle F12511 as a promising therapeutic approach for targeting ACAT inhibition in the brain.”

K604 is a well-known ACAT1 specific inhibitor that has been extensively employed for ACAT1 inhibition (Shibuya et al., reference in line 182). However, our experimental results have clearly demonstrated that nanoparticle F exhibits remarkable ACAT1 inhibitory activity in the brain within 4 hours, whereas nanoparticle K604 did not yield similar effects (Figure 1D). Based on these compelling findings, we have made the decision to proceed with further investigations focusing on nanoparticle F. This decision stems from its promising potential as a superior candidate for our study, given its demonstrated efficacy in ACAT1 inhibition within the brain.

Then, these NPs are mentioned in the results (only in the first part), but there is no mention of treating the animals with those in the methods. 

Thank you for your comment. We appreciate your attention to detail. We have taken your suggestion into consideration and have included additional information to clarify our animal treatment method. In the revised manuscript, in Section 4.1 (Ethical handling of animals), we have added the following statement to provide further clarification:

"Depending on the specific assay being performed, mice were either administered with nanoparticle F at doses of 5.8 mg/kg or 46 mg/kg, or with nanoparticle K604 at a dose of 40 mg/kg." (Line 789-790)

This addition aims to provide a clear and concise description of the animal treatment method used in our study.

Also, in lines 474-489: the authors described an assay using mouse tissues. However, it is not clear from which animals were the tissues extracted. It is from the mice used in this work? And, which tissues were used? 

Thank you for bringing up this question. We appreciate your attention to detail. In our study, the tissue samples used were obtained from wild-type mice (C57BL/6). These samples included the forebrain, cerebellum/brain stem, adrenal gland, and liver. We apologize for any confusion caused by the omission of this information in the original manuscript. We have now updated the method section to include this important detail.

Furthermore, drug delivery to the brain is very challenging and hinders the success of many therapeutic drugs. However, no active targeting strategies were explored by the authors.  

Thank you for your suggestion. We appreciate your insight and agree that exploring active targeting strategies is a valuable direction for future research. We have now included a discussion on active targeting strategies in our future work section (Line 826-834).

Round 2

Reviewer 1 Report

The authors have addressed my concerns on missing data and F12511 control experiment.  However, do the authors intend to include S1 and S2 in the manuscript or should the authors upload a new supplementary file with the gavage data (S1). 

"Supplementary Materials: The following supporting information can be downloaded at: 1010 www.mdpi.com/xxx/s1, Figure S1: Brain cytokines differentiate upon treatment with DSPE-PEG/PC 1011 and nanoparticle F.", needs to be updated to include gavage data if being included in supplementary.

Also, while this is the authors decision, there seems to be a disconnect in figure formatting (color, text size, points/characters for each value, statistical connecting line size).  

Importantly, please include a statement on statistical analysis in the methods.  Also, specific statistical test used should be included in all figure legends that denote significant differences between treatments.  For example, Figure 7 includes "One-way ANOVA was conducted for statistics. N.S. not significant; p<0.001 ***, p<0.01 **,p<0.05 *.", while figure 3 has no information.

Author Response

Reviewer #1- Comments and Suggestions for Authors 

  1. The authors have addressed my concerns on missing data and F12511 control experiment.  However, do the authors intend to include S1 and S2 in the manuscript or should the authors upload a new supplementary file with the gavage data (S1). 

       "Supplementary Materials: The following supporting information can be downloaded at: 1010 www.mdpi.com/xxx/s1, Figure S1: Brain cytokines differentiate upon treatment with DSPE-PEG/PC 1011 and nanoparticle F.", needs to be updated to include gavage data if being included in supplementary.

       We appreciate the reviewer’s positive feedback. Thank you very much for your suggestion, we have uploaded both S1 and S2 in the Supplementary file. We also updated the Supplementary Materials section accordingly (line 745-748).

  1. Also, while this is the authors decision, there seems to be a disconnect in figure formatting (color, text size, points/characters for each value, statistical connecting line size).  

       Importantly, please include a statement on statistical analysis in the methods.  Also, specific statistical test used should be included in all figure legends that denote significant differences between treatments.  For example, Figure 7 includes "One-way ANOVA was conducted for statistics. N.S. not significant; p<0.001 ***, p<0.01 **,p<0.05 *.", while figure 3 has no information.

       Thank you for your comments, we have reviewed the figures, and made the necessary corrections to both Figure 1 and Figure 3.

Reviewer 4 Report

The new version of the manuscript has been much improved, figures have been put in order and corrected. However there are still some items that should be improved. Some are listed below:

 -       Page 4, line 174, “the inhibition does not become diluted by washing the cells,” an inhibition effect can decrease but not be diluted or became diluted. The sentence should be changed with “the inhibition effect does not decrease by washing the cells”;

-    -  Page 4, lines 193-196, “Overall, our result demonstrated that, in our nanoparticle system, F12511 is superior in inhibiting ACAT in the brain compared to K604. Therefore, the remaining of our study will focus on understanding nanoparticle F pharmacokinetics, biodistribution & efficacy in the brain”, what author mean with “is superior”? The sentence should be changed with “Overall, our result demonstrated that, in our nanoparticle system, the effect of F12511 in inhibiting ACAT in the brain is higher compared to K604. Therefore, the remaining of our study will focus on understanding nanoparticle F pharmacokinetics, biodistribution and efficacy in the brain”;

-      -   Figure 1’s caption, what 5 mol% means?, what units are mol%?

-      -   Figures’ caption should not content any reference, e.g. Figure 3, 5, 6  ; this information should be included in the results section;

-      -    Figure 4, scale bars information is missed in the caption;

-    -     Page 16, line 808, change “poses” with “possesses”;

-     -    Page 19, line 981, section 2.5 does not content the information to prepare brain homogenates; In section 4.11 there are several important mistakes.  

-   -   Authors wrongly use the word “spike”, perhaps they mean “positive/high-  and  low-cytokine values. A spike sample is made of the buffer in which brain homogenates have been prepared and the biomarker/s to a known concentration in order to investigate the “matrix effect” in the quantification. This should be done, in particular when the sample is tissue using as standards diluent the buffer used for the homogenization. This is the “recovery”, not what authors have described as “recovery”.   

Overall, figures style should be harmonized through all the manuscript. Some figures have very big size characters whereas others are too small, e.g. Figure 3 and 4 compared with Figure 1, 2.

There are some mistakes in the manuscript, they have been listed in the "Comments and Suggestions for authors and Editor"

Author Response

Reviewer #4- Comments and Suggestions for Authors 

The new version of the manuscript has been much improved, figures have been put in order and corrected. However there are still some items that should be improved. Some are listed below:

1. Page 4, line 174, “the inhibition does not become diluted by washing the cells,” an inhibition effect can decrease but not be diluted or became diluted. The sentence should be changed with “the inhibition effect does not decrease by washing the cells”;

Thank you for your positive comment.

We have made the correction to the following sentences, in lines 146-147.

“The inhibition of the ACAT enzyme by F12511 or K604 remains unaffected by cell washing or the cell homogenization preparation process.”

2Page 4, lines 193-196, “Overall, our result demonstrated that, in our nanoparticle system, F12511 is superior in inhibiting ACAT in the brain compared to K604. Therefore, the remaining of our study will focus on understanding nanoparticle F pharmacokinetics, biodistribution & efficacy in the brain”, what author mean with “is superior”? The sentence should be changed with “Overall, our result demonstrated that, in our nanoparticle system, the effect of F12511 in inhibiting ACAT in the brain is higher compared to K604. Therefore, the remaining of our study will focus on understanding nanoparticle F pharmacokinetics, biodistribution and efficacy in the brain”;

We have edited our manuscript as follows, in lines 165-169.

Overall, our results demonstrate that, in our nanoparticle system, F12511 exhibits significantly stronger inhibitory effects on ACAT activities in the mouse brain compared to K604. Therefore, the subsequent focus of our study will be on understanding the pharmacokinetics, biodistribution and efficacy of nanoparticle F in the brain.”

3. Figure 1’s caption, what 5 mol% means? what units are mol%?

If  a mixture consists of 10 moles of component A, 20 moles of component B, and 15 moles of component C. The total number of moles in this mixture is 10+20+15=45. We can then report that mol% of component A=10 moles/45 moles *100= 22.22%.

In our nanoparticle system: DEPE-PEG is 30 mM, PC is 6 mM, and F12511 at high dose of 46 mg/kg of mouse weight, and low dose of 5.8 mg/kg of mouse weight. Then the total number of molecules in our nanoparticles is 30+6+12=48, at high dose; or 30+6+1.51=37.51 moles at low dose. Therefore, the mole% of F12511 is 12/48*100=25 mole% at high dose; or 1.51/37.5*100=4.03 mol%.

To avoid confusion, now we revised our manuscript, in the Materials and Methods section, at the 4.3 Nanoparticle formation(Line 606-614), we revised to the following paragraph:

In the initial studies, DSPE-PEG2000 nanoparticles were employed, loaded with F12511 at varying concentrations. The concentration ranged from a low dose of 1.51 mM (4 mol%; applied to the mouse at 5.8 mg F per kg body weight) to a high dose of 12 mM (25 mol%; applied to mouse at 46 mg F per kg body weight). The nanoparticles underwent the same lyophilization procedure. Subsequently, after re-solubilization in 1 mL PBS, the nanoparticles were probe-sonicated on ice under sterile condition, using a Branson probe-sonicator. The sonication procedure consisted of two sets of  1-min pulses with 5-min rest period between sets. The resulting nanoparticles were then used for injection.”

And at the legend of Figure 1, Lines 171-175, we changed to:

WT mice were IV injected with either nanoparticle F with F12511 at low concentration (30 mM DSPE-PEG2000 with 5.8 mg F per kg mouse body-weight), or with empty nanoparticle at zero time and sacrificed after (A) 1 h, (B) 4 h, (C) 8 h. (D) WT mice were IV injected with nanoparticle K604, with K604 at high concentration (30 mM DSPE-PEG2000 with 24 mg K604 per kg), or with empty nanoparticle at zero time and sacrificed after 4 h.”

4. Figures’ caption should not content any reference, e.g. Figure 3, 5, 6  ; this information should be included in the results section;

Thank you for bringing this to our attention. We have deleted the references in Figures 1, 3, 5, 6.

5. Figure 4, scale bars information is missed in the caption.

We have added the following message in the legend section of Figure 4, Line 281-282.

Scale bars: 200 pixels: 109 µm for 10x enlargement; and ~54 µm for 20x enlargement.

6. Page 16, line 808, change “poses” with “possesses”;

Thank you for pointing out the typo. It has been corrected at Line 564.

7. Page 19, line 981, section 2.5 does not content the information to prepare brain homogenates.  

Thank you for pointing out the mistake, this should refer to section 4.7

"Preparation of brain homogenates and Western blot analysis"

8. In section 4.11 there are several important mistakes

Authors wrongly use the word “spike”, perhaps they mean “positive/high-  and  low-cytokine values. A spike sample is made of the buffer in which brain homogenates have been prepared and the biomarker/s to a known concentration in order to investigate the “matrix effect” in the quantification. This should be done, in particular when the sample is tissue using as standards diluent the buffer used for the homogenization. This is the “recovery”, not what authors have described as “recovery”.   

Thank you for your comment. We agree with your feedback. We change “spike” to “quality control sample”. To avoid any confusion, we have now modified this section to the following:

  1. High- and low-quality control samples with a known concentration, provided by the manufacturer, were utilized to validate the calculation of the standard curve. Standards and quality control samples were measured in triplicate, while samples were measured once, and blank values were subtracted from all readings to ensure accurate measurement. (Line 718-722).
  2. Briefly, each well was pre-wet with 100 µL PBS containing 1% BSA. Then, beads, along with a standard, sample, quality control samples, or blank, were added in a final volume of 100 µL. The plate was incubated at room temperature for 30 min with continuous shaking.” (Line 724-727).
  1. Overall, figures style should be harmonized through all the manuscript. Some figures have very big size characters whereas others are too small, e.g. Figure 3 and 4 compared with Figure 1, 2.

 Thank you for bringing it to our attention. We have now made the necessary adjustments to decrease the character size in Figures 3 & 4.

Reviewer 5 Report

The authors replied to my comments and made the requested changes 

Author Response

We appreciate the reviewer’s positive comment.